# Predictive Validity, Diagnostic Accuracy and Test-Retest Reliability of the Strength of Urges to Drink (SUTD) Scale

**DOI:** 10.3390/ijerph16193714

**Published:** 2019-10-02

**Authors:** Emma Beard, Jamie Brown, Robert West, Colin Drummond, Eileen Kaner, Susan Michie

**Affiliations:** 1Department of Behavioural Science and Health, University College London, London WC1E 7HB, UK; jamie.brown@ucl.ac.uk (J.B.); robert.west@ucl.ac.uk (R.W.); 2Department of Clinical, Educational and Health Psychology, University College London, London WC1E 7HB, UK; s.michie@ucl.ac.uk; 3National Addiction Centre, Institute of Psychiatry, Psychology & Neuroscience, King’s College London, London SE5 8BB, UK; colin.drummond@kcl.ac.uk; 4Institute of Health & Society, Newcastle University, Newcastle upon Tyne NE2 4AX, UK; eileen.kaner@newcastle.ac.uk

**Keywords:** strength of urges to drink (SUTD), validity, reliability, AUDIT

## Abstract

This study compared the 1-item Strength of Urges to Drink (SUTD) scale with the 10-item Alcohol Use Disorders Identification Test (AUDIT) on (i) test-retest reliability, (ii) predictive validity, and (iii) diagnostic accuracy. Data come from 2960 participants taking part in the Alcohol Toolkit Study (ATS), a monthly population survey of adults in England. The long-term test-retest reliability of the SUTD was ‘fair’, but lower than that for the AUDIT (Kappa_weighted_ 0.24 versus 0.49). Individuals with “slight/moderate” urges to drink had higher odds of reporting an attempt to cut down relative to those not experiencing urges (adjusted odds ratios (AdjORs) 1.78 95% confidence interval (CI) 1.43–2.22 and 1.54 95% CI 1.20–1.96). Drinkers reporting “moderate/slight/strong” urges to drink had mean change in consumption scores which were 0.16 (95% CI −0.31 to −0.02), 0.40 (95% CI −0.56 to −0.24) and 0.37 (95% CI −0.69 to −0.05) units lower than those reporting no urges. For all outcomes, strong associations were found with AUDIT scores. The accuracy of the SUTD for discriminating between drinkers who did and did not reduce their consumption was ‘acceptable’, and similar to that for the AUDIT (ROC_AUC_ 0.6). The AUDIT had better diagnostic accuracy in predicting change in alcohol consumption. The SUTD may be an efficient dynamic measure of urges to drink for population surveys and studies assessing the impact of alcohol-reduction interventions.

## 1. Introduction

Worldwide each year around 6 L on average of pure alcohol are consumed by every person aged 15 years or older [1]. A large variation exists in adult per capita consumption with the highest consumption levels found in the developed world. In England, around 17% (~9 million) of adults drink alcohol above recommended limits [2] and 6% (~1 million) of the population are classified as dependent i.e., they have a physical and/or mental dependency on alcohol which is associated with high levels of tolerance to its effects and withdrawal symptoms when absent [3]. Such consumption levels are associated with a number of non-communicable diseases, injury and alcohol attributable death each year [1].

The reliable measurement of the severity of alcohol dependence is important for several reasons. First, it can be helpful when deciding how much and what kind of support should be offered. Currently, less than 14% of dependent drinkers receive mental or emotional support [3,4]. Secondly, it is useful in epidemiological studies to be able to characterise the dependency of the population of drinkers. This may be for descriptive purposes or to allow comparisons between those who do and do not drink excessively. 

The diagnosis of an alcohol use disorder has traditionally been based primarily on the findings of an interview using either the International Statistical Classification of Diseases and Related Health Problems (ICD-10) [5] or the Diagnostic and Statistical Manual of Mental Disorders (DSM-V) [6] as diagnostic instruments. Questionnaires have also been devised, with the most widely used screening tool known as the Alcohol Use Disorders Identification Test (AUDIT). The AUDIT measures harmful and hazardous drinking, and possible dependency. Whereas hazardous drinking is often defined as a quantity or pattern of alcohol use that places individuals at risk of adverse health events, harmful consumption is defined as alcohol intake which results in actual physical or psychological harm [7]. Since alcohol dependence is a construct which is hard to define in terms of any single measure, the AUDIT consists of 10 questions assessing frequency of drinking, average amount consumed, concerns from others, harm to self and others, inability to function without alcohol and alcohol-induced amnesia [8]. Although numerous studies have demonstrated this questionnaire’s reliability and validity across socio-economic groups and cultures [9,10,11,12,13], its use is somewhat limited by its length and complexity. 

How best to characterise alcohol dependence is a continuing debate. Key features adopted by the ICD-10 include an inner drive or compulsion to consume alcohol, continued drinking despite harm and commonly a withdrawal state upon stopping drinking [14,15,16,17]. This conceptualisation of dependence raises the possibility of a simpler measure in terms of urges to drink, where urges can be seen as an emotional state that is characterised by the motivation to seek and use alcohol [18]. PRIME theory (which represents the motivation system consisting of plans, responses, impulses, motives and evaluations) provides a relevant framework [19]. According to PRIME theory, we act every moment in pursuit of what we most *want* or *need*. These wants and needs influence behaviour through momentary impulses and inhibitions, and can be felt as urges to perform the behaviour e.g., to drink. Strength of urges appear to be a useful measure of cigarette addiction, as are the related measures of craving and motivation to smoke [20,21]. Given evidence of a similar underlying biological pathway for the two behaviours, it is hypothesised that urges to drink could also be a valid measure of alcohol dependence [22]. Indeed, drugs which attenuate urges to drink have been associated with reduced consumption [23], while findings suggest that experiencing strong urges is related to consuming more alcohol and associated with use that is harmful and hazardous [24].

Several questionnaires are available that assess urges and cravings to drink. For example, the Alcohol Urges Questionnaire measures acute urges and comprises eight items pertaining to desire to drink, expectations of positive effects from drinking and inability to avoid alcohol [25]. The Alcohol Craving Questionnaire contains 47 items covering five domains: desire to drink, intention to drink, lack of control, anticipation of positive effects and expectancy of relief from withdrawal [26], while the Yale-Brown Obsessive Compulsive Scale for Heavy Drinking consists of 10 items with obsessionality and compulsive subscales [27]. A popular questionnaire known as the Severity of Alcohol Dependence Questionnaire (SADQ) [28] is a self-administered 20 item questionnaire with five subscales measuring dependency: Physical Withdrawal, Affective Withdrawal, Withdrawal Relief Drinking, Alcohol Consumption, and Rapidity of Reinstatement. Whilst these have been validated, many are not widely used in the alcohol arena, and some, including the AUDIT and SADQ, measure recent or past alcohol use and related feelings which may limit their use when looking at more chronic aspects of dependence which are transient [29]. There is a need for a simpler measure which captures the dynamic nature of urges to drink and is applicable to both dependent drinkers and those consuming alcohol at harmful levels. Drinking occurs on a continuum, and hence it is useful to be able to identify harmful drinkers who may be around the threshold for dependence and, therefore, tertiary preventive work can be used to help stop further escalation of problems.

Thus, this study aimed to evaluate psychometric properties of the strength of urges to drink on a single day, known as the Strength of Urges to Drink (SUTD) measure, among a population sample of high-risk drinkers. Such epidemiological data has several advantages over patient populations, including the fact that many individuals who are alcohol-dependent remain undiagnosed. Population-based studies may be able to pick some of these individuals up [30,31]. Comparisons will be made with the AUDIT, as it is the most widely used screening tool and can be self-completed. 

More specifically, it aimed to assess the:Test-retest reliability of the SUTD compared to the AUDIT.Predictive validity of the SUTD compared to the AUDIT in relation to (a) reported attempts to reduce alcohol consumption between baseline and follow-up; (b) reported alcohol consumption at follow-up and (c) change in alcohol consumption between baseline and follow-up.Diagnostic accuracy of the SUTD compared to the AUDIT in relation to (a) attempts to reduce alcohol consumption between baseline and follow-up; (b) alcohol consumption at follow-up and (c) change in alcohol consumption between baseline and follow-up.

## 2. Methods

### 2.1. Design and Setting

Data were used from repeated cross-sectional household surveys of a representative sample of the population of adults in England conducted in consecutive monthly waves between March 2014 and December 2016. The surveys are part of the ongoing Alcohol Toolkit Study which is designed to provide tracking information about alcohol consumption and related behaviours in England. Each month a new sample of approximately 1700 adults aged 16+ complete face-to-face computer assisted interviews. All respondents are asked if they are happy to be re-contacted 6 months after baseline [32]. The baseline survey uses a type of random location sampling, which is a hybrid between random probability and simple quota sampling. England is first split into 171,356 ‘Output Areas’, comprising approximately 300 households. These areas are then stratified based on ACORN characteristics and geographic region. ACORN (A Classification Of Residential Neighbourhoods) is a socio-economic profiling tool developed by Acorn Consumer Classification (CACI) [33]. The areas are then randomly allocated to interviewers, who travel to their selected areas and conduct the electronic interviews with one member of the household. Interviews are conducted until quotas based upon factors influencing the probability of being at home and tailored to local area census data are fulfilled. Morning interviews are avoided to maximise participant availability.

### 2.2. Participants

Data were collected on 57,341 participants over the study period. Of these, 27.02% (95% confidence interval (CI) 26.32 to 27.71, *n* = 15,492; unweighted: 25.53%; 95% CI 24.82 to 26.24, *n* = 14,639) were high-risk drinkers and form the sample for this study.

### 2.3. Ethical Approval

Ethical approval for the Smoking Toolkit Study (STS), a sister survey to the Alcohol Toolkit Study (ATS), was originally granted by the UCL Ethics Committee (ID 0498/001). Approval for the ATS was granted by the same committee as an extension of the STS.

### 2.4. Measures

At baseline, participants were asked questions that assessed: age; sex; an occupationally-based classification of socio-economic status called ‘social grade’ (dichotomised to ABC1  =  higher and intermediate professional/managerial and supervisory, clerical, junior managerial/administrative/professional or C2DE  =  skilled, semi-skilled, unskilled manual and lowest grade workers or unemployed); government office region in England (dichotomised to North  =  North East, North West, and Yorkshire and the Humber, East Midlands, West Midlands, or South  =  East of England, London, South East, and South West, classified according to an established North–South divide); receipt of a voluntary educational qualification (obtained after compulsory education ceases at 16 years old); ethnicity (dichotomised as white versus other); and disability. They were also asked if they were currently attempting to cut down on their alcohol consumption.

Participants were also asked to complete the AUDIT questionnaire [10,34,35] and the SUTD measure which consists of one item: “How strongly have you felt the urge to drink in the past 24 h?” Responses include: not at all, slight, moderate, strong, very strong and extremely strong.

The AUDIT-Quantity/Frequency scale (AUDIT-QF) [36] comprises the first two questions of the AUDIT:“How often do you have a drink containing alcohol?” Responses include: never, monthly or less, 2–4 times a month, 2–3 times a week and 4+ times a week.“How many units of alcohol do you drink on a typical day when you are drinking?” Responses include: 1–2 drinks, 3–4 drinks, 5–6 drinks, 7–9 drinks and 10+ drinkers.

Scores on these two questions are combined to give a measure of alcohol consumption, with a range of 0 to 8. 

Those who scored 8 or more (i.e., indicating hazardous and or harmful alcohol consumption and possible dependence) on the AUDIT or 5 or more on the AUDIT-C, which comprises the first three questions of the AUDIT, (i.e., indicating high-risk consumption) at baseline were then re-contacted at 6-months follow-up and asked to complete the SUTD, AUDIT and AUDIT-QF questionnaires and: “How many attempts to restrict your alcohol consumption have you made in the last 6 months (e.g., by drinking less, choosing lower strength alcohol or using smaller glasses)?

### 2.5. Analyses

The protocol for this study was published on the Open Science Framework prior to data analysis (https://osf.io/wuuqr/). An amendment was made to the analysis plan in February 2017: we added a plan to assess the predictive validity of the SUTD in relation to the *change* in consumption between baseline and follow-up.

All analyses were conducted in R version 3.3.2. Data were weighted for key prevalence statistics (for more details see [32]). Those who were and were not followed up were compared on key baseline variables to establish representativeness of the follow-up sample using Mann–Whitney U, *t*-tests and chi-square tests as appropriate.

Test-retest reliability was assessed by calculating: (a) a reliability coefficient (r), which is simply the Spearman’s correlation between the scores on the first and the second testing. The value for the r coefficient can fall between 0.00 (no correlation) and 1.00 (perfect correlation); and (b) a weighted kappa coefficient which is suitable for ordinal data. Values can range from −1 to 1, where 1 indicates perfect agreement, 0 indicates no agreement beyond chance and negative values indicate inverse agreement. Cohen suggested the Kappa result be interpreted as follows: values ≤0 as indicating no agreement and 0.01–0.20 as none to slight, 0.21–0.40 as fair, 0.41–0.60 as moderate, 0.61–0.80 as substantial, and 0.81–1.00 as almost perfect agreement [37].

The predictive validity of the SUTD was evaluated by examining the association between the SUTD scale and (a) attempts to reduce alcohol intake; (b) levels of alcohol consumption at follow-up and (c) change in alcohol consumption between baseline and follow-up using a Mann–Whitney U test and linear-by-linear association chi-square test. Next attempts to reduce alcohol intake, levels of alcohol consumption at follow-up and change in alcohol consumption between baseline and follow-up were regressed on to the baseline scores using simple logistic and linear regression and multiple logistic and linear regression, adjusting for the following covariates measured at baseline: age, sex, social grade, region, receipt of a voluntary educational qualification, ethnicity, disability, AUDIT and wave of the survey.

To assess predictive accuracy, Received Operating Characteristic (ROC) curves were then calculated [38]. The ROC curve is a graphical presentation of the accuracy of a measure in which the sensitivity of the measure (i.e., the true positive rate) is plotted against the specificity (i.e., the false positive rate). The area under the ROC curve (ROC_AUC_) has a value from 0.5 (chance level only) to 1 (perfect discrimination). Alcohol consumption and change in alcohol consumption were first dichotomised according to their mean value into lower and higher scores [39]. Our a priori hypothesis was that the SUTD would be as accurate in discriminating whether drinkers attempt to cut down at follow-up and whether drinkers have a lower or higher alcohol consumption than the mean.

After viewing the distributions and associations for the SUTD additional unplanned sensitivity analyses were run collapsing the “moderate”, “strong”, “very strong” and “extremely strong” categorises into a three item SUTD scale (SUTD-3) comprising of “not at all”, “slight” and “moderate > 3”. Unplanned analyses were also run to assess the predictive and diagnostic accuracy of the SUTD, SUTD-3 and AUDIT in relation to consumption at follow-up and a change in consumption between baseline and follow-up restricted to those making an attempt to cut down at baseline. This analysis more accurately mirrors the previous association established between the Strength of Urges to Smoke (SUTS) scale and the success of attempts to quit smoking [20].

Strengthening The Reporting of OBservational studies in Epidemiology (STROBE) guidelines for the reporting of observational epidemiological studies were followed throughout [40].

## 3. Results

The sample followed up 6 months after baseline (*n* = 2960) differed from those not followed up (*n* = 11,679). They were more likely to be older, to report currently attempting to cut down their alcohol consumption, to be from high socio-economic status, to have a disability, to reside in the South of England, to have stronger urges to drink and to have higher AUDIT scores (Table 1).

Figure 1 shows the distribution of scores on the SUTD measure at baseline and follow-up. At baseline the two most frequently reported categories were “not at all” and “slight”. Nineteen per cent (*n* = 2730) and 11.9% (*n* = 352) scored in the highest four categories (i.e., greater than moderate at baseline and follow-up, respectively.

### 3.1. Test-Retest Reliability

In terms of test-retest reliability of the SUTD, scores at baseline and 6-month follow-up correlated weakly, r = 0.30 (95% CI 0.28 to 0.34) and weighted kappa suggested ‘fair’ reliability (wk = 0.24, 95% CI 0.28 to 0.32). The full AUDIT had slightly better test re-test reliability (r = 0.50, 95% CI 0.47 to 0.53; wk = 0.49; 95% CI 0.53 to 0.57). Test re-test reliability for the SUTD-3 was similar (r = 0.30, 95% CI 0.27 to 0.33; wk = 0.26, 95% CI 0.30 to 0.34).

### 3.2. Predictive Validity

#### 3.2.1. Attempts to Cut Down between Baseline and 6-Month Follow-Up

A total of 767 higher risk drinkers (25.9%; 95% CI 24.3 to 27.5) reported that they had attempted to reduce their alcohol consumption between baseline and follow-up.

Table 2 presents the percentage of high-risk drinkers reporting an attempt to cut down stratified by their baseline SUTD score. For the full SUTD measure there is a clear monotonic relationship between the percentage attempting to cut down and increasing urges to drink (U = 974,080, *p* < 0.001). Of the 23 drinkers who scored the two highest levels of urges to drink, 37.1% had attempted to cut down. The relationship is also monotonic for the SUTD-3 (U = 973,570, *p* < 0.001).

The odds of attempting to cut down between baseline and the 6-month follow-up according to the SUTD and SUTD-3 scales are also presented in Table 2. For the SUTD, drinkers reporting “slight” to “very strong” urges to drink had 1.57 to 1.09 times higher odds of making an attempt to cut down than drinkers who reported “not at all”. After adjusting for age, sex, social grade, region, receipt of a voluntary educational qualification, ethnicity, disability, AUDIT scores and wave of the survey, those reporting “slight” and “moderate” urges had 1.78 and 1.58 higher odds of an attempt to cut down. For the SUTD-3 scale, those reporting “slight” and “moderate >” urges to drink had a 1.78 and 1.51 higher odds of reporting an attempt to cut down at follow-up in adjusted analyses.

In comparison, a positive association was also found between AUDIT scores and attempts to cut down in unadjusted analyses (odds ratio (OR) 1.09, 95% CI 1.07 to 1.12, *p* < 0.001). This significant association remained after adjustment for all other variables in Table 2 (OR 1.10; 95% CI 1.07 to 1.12, *p* < 0.001).

#### 3.2.2. Reported Alcohol Consumption at 6-Month Follow-Up

##### All Participants

The mean consumption score at follow-up was 4.7 (95% CI 4.6 to 4.7). Table 2 presents the mean consumption scores of high-risk drinkers stratified by their baseline SUTD and SUTD-3 scores. There appears to be an almost linear increase in mean consumption scores with increasing urges to drink on SUTD score (z = 9.821, *p* < 0.001) and SUTD-3 scale (z = 10.533, *p* < 0.001). Of the 26 drinkers who scored the two highest levels of urges to drink on the SUTD, the mean consumption score was 5.14.

Drinkers reporting “moderate”, “strong”, “very strong”, and “extremely strong” urges to drink on the SUTD had mean consumption scores which were 0.30, 0.77, 0.80 and 0.56 units higher than those reporting “not at all” (Table 2). The beta values were smaller after adjusting for age, sex, social grade, region, receipt of a voluntary educational qualification, ethnicity, disability, AUDIT scores and wave of the survey. On the SUTD-3 those reporting “slight” and “>moderate” urges had consumption scores which were 0.22 and 0.46 units higher than those not experiencing urges to drink in adjusted analyses.

By comparison, in unadjusted analyses, a positive association was found between AUDIT scores and consumption levels (β 0.13, 95% CI 0.11 to 0.14, *p* < 0.001). This significant association remained after adjustment for all other variables in Table 2 (β 0.13; 95% CI 0.08 to 0.35, *p* < 0.001).

##### Participants Cutting Down at Baseline

Table 3 presents the mean consumption scores of high-risk drinkers who reported cutting down at baseline stratified by their baseline SUTD and SUTD-3 scores. The mean consumption score at follow-up among those cutting down at baseline (*n* = 692) was 4.9 (95% CI 4.8 to 4.9). There appeared to be an almost linear increase in mean consumption scores with increasing urges to drink on the SUTD (z = 2.5733, *p* = 0.010) and SUTD-3 (z = 3.3679, *p* < 0.001).

In unadjusted analyses, the data were inconclusive as to whether those reporting “slight”, “strong”, “very strong” and “extremely strong” urges to drink on the SUTD had different consumption levels at follow-up relative to those reporting “not at all”. In contrast, those with “moderate” urges to drink had significantly higher consumption levels (Table 3). For the SUTD-3, the data were inconclusive as to whether those reporting “slight” urges to drink had different consumption levels at follow-up relative to those reporting “not at all”. In contrast, those reporting “>moderate” urges had significantly higher consumption levels in unadjusted but not adjusted analyses.

A positive association was found for the AUDIT scores and consumption levels in both adjusted and unadjusted analyses (β 0.08; 95% CI 0.05 to 0.10, *p* < 0.001 versus β_adj_ 0.09; 95% CI −0.07 to 0.12, *p* < 0.001).

#### 3.2.3. Change in Alcohol Consumption between Baseline and 6-Month Follow-Up

##### All Participants

The mean change in consumption scores between baseline and follow-up was 0.32 (95% CI 0.30 to 0.36). Table 2 presents the change scores stratified by their baseline SUTD score and shows a non-linear association for the full SUTD scale (z = −1.7804, *p* = 0.075) and SUTD-3 scale (z = −3.3012, *p* < 0.001). Of the 26 drinkers who scored the two highest levels of urges to drink on the SUTD scale, the mean change in consumption score was 0.86.

Drinkers reporting “moderate” and “extremely strong” urges to drink on the SUTD had mean change in consumption scores which were 0.3 units lower and 1.05 units higher than those reporting “not at all” (Table 2). The differences were smaller after adjusting for age, sex, social grade, region, receipt of a voluntary educational qualification, ethnicity, disability, AUDIT scores and wave of the survey. For the SUTD-3 changes in consumption were smaller for those in the “slight” and “moderate >” relative to those not reporting urges to drink after adjustment.

By comparison, in unadjusted analyses, a positive association was found between AUDIT scores and the change in consumption levels (β 0.05, 95% CI 0.04 to 0.07, *p* < 0.001). This significant association remained after adjustment for all other variables in Table 2 (β 0.03; 95% CI 0.05 to 0.08, *p* < 0.001).

##### Participants Cutting Down at Baseline

The mean change in consumption among those currently cutting down at baseline (*n* = 692) was 0.20 (95% CI 0.08 to 0.32). Table 3 presents the mean change in consumption scores of high-risk drinkers who reported cutting down at baseline stratified by their baseline SUTD and SUTD-3 scores. There was no linear association between mean consumption scores and urges to drink on the SUTD (z = 1.4292, *p* = 0.153) or SUTD-3 (z = 0.065, *p* = −0.9481).

In unadjusted analyses, the data were inconclusive as to whether those reporting “slight”, “moderate”, “strong” and “very strong” urges to drink on the SUTD had different consumption change scores relative to those reporting “not at all” (Table 3). In contrast, those with “extremely strong” urges to drink had significantly larger change scores, suggesting a significantly larger increase in consumption between baseline and follow-up. For the SUTD-3, the data were inconclusive as to whether those reporting “slight” and “>moderate” urges to drink had different consumption change scores to those reporting “not at all”.

In contrast, a positive association was found for the AUDIT scores and the change in consumption levels in both adjusted and unadjusted analyses (β 0.07; 95% CI 0.05 to 0.10, *p* < 0.001 versus β_adj_ −0.06; 95% CI −0.03 to 0.09, *p* < 0.001).

### 3.3. Diagnostic Accuracy

Figure 2a shows the ROC curve for the six-item SUTD measure predicting attempts to cut down. The ROC_AUC_ was 0.6 (95% CI 0.5 to 0.6). The ROC_AUC_ for the AUDIT (0.6; 95% CI 0.6 to 0.7) and SUTD-3 (0.6; 95% CI 0.5 to 0.6) were similar. This would suggest that scores on the SUTD, AUDIT and SUTD-3 would lead to correct categorisation of whether one will make an attempt to cut down around 60% of the time.

Figure 2b shows the ROC curve for the six-item SUTD measure predicting consumption levels at follow-up. The ROC_AUC_ was 0.6 (95% CI 0.5 to 0.6). The ROC_AUC_ for the AUDIT (0.7; 95% CI 0.6 to 0.7) but for the SUTD-3 (0.6; 95% CI 0.5 to 0.6) was similar. This would suggest that scores on the SUTD and SUTD-3 would lead to correct categorisation of consumption around 60% and on the AUDIT around 70% of the time. When restricting the analysis to those cutting down at baseline, the ROC_AUC’s_ were as follows: SUTD (0.5; 95% CI 0.5 to 0.6), SUTD-3 (0.5; 95% CI 0.4 to 0.6) and AUDIT (0.7; 95% CI 0.6 to 0.7).

Figure 2c shows the ROC curve for the six-item SUTD measure predicting change in consumption levels between baseline and follow-up. The ROC_AUC_ was 0.5 (95% CI 0.5 to 0.6). The ROC_AUC_ for the AUDIT was slightly higher (0.6; 95% CI 0.5 to 0.6) but for the SUTD-3 (0.5; 95% CI 0.5 to 0.6) was similar. This would suggest that scores on the SUTD, SUTD-3 and AUDIT would lead to correct categorisation of consumption around 50% and 60% of the time. When restricting the analysis to those cutting down at baseline, the ROC_AUC’s_ were as follows: SUTD (0.6; 95% CI 0.5 to 0.6), SUTD-3 (0.6; 95% CI 0.5 to 0.7) and AUDIT (0.6; 95% CI 0.5 to 0.7).

## 4. Discussion

Although the long-term test re-test reliability was better for the AUDIT it was still fair for the SUTD [37]. The SUTD was associated with heavier alcohol consumption at follow-up, a reduction in alcohol consumption between baseline and follow-up and greater likelihood of attempting to cut down between baseline and 6-months follow-up. The accuracy of the SUTD in discriminating between drinkers who attempted to reduce and did not attempt to reduce their alcohol intake, and drinkers with a consumption level lower than the mean and higher than the mean consumption level at follow-up, was around 0.6 which would be broadly considered as acceptable [41]. However, the SUTD was poor at discriminating between those with a change in consumption level between baseline and follow-up which was lower than the mean and higher than the mean. The AUDIT had acceptable to good discriminatory accuracy for all outcomes, performing better than the SUTD on predicting consumption levels and change in consumption levels.

This study has several advantages including the use of data from a large household survey of adults in England which enabled the assessment of the validity and reliability of the SUTD scale compared to the widely-validated AUDIT questionnaire. However, this study also has several limitations which must be considered. First, is the low response rate at 6-months follow-up which may have introduced bias. However, differences between those followed and those not followed up were small. Secondly, participants were asked to retrospectively recall attempts to cut down on their alcohol intake and thus it is possible attempts were forgotten. This may have led to an underestimation of the association with urges to drink. Thirdly, this paper assessed the association between SUTD measures and changes in alcohol consumption at one time point. Given the transient nature of urges to drink it will be important to assess in further studies associations using repeated longitudinal measures (e.g., ecological momentary assessment) and also the relationship with other outcomes including relapse. Finally, interviews could happen at any time of the day but morning interviews were avoided to maximise availability. It is possible that urges to drink are different in the morning and evening and that any differences are moderated by dependence levels. Dependent drinkers may have a greater urge for ‘relief drinking’ in the morning, while heavy non-dependent drinkers could be more affected by cues for early evening drinking.

Test-retest reliability of AUDIT scores has been shown to be high at least in the short term (e.g., r > 0.6) [42]. The poorer reliability of the SUTD identified in this study may reflect the gap of 6 months between measurement periods, with the AUDIT questionnaire assessing dependency over the past few weeks, while the SUTD measures urges over the past 24 h. Lack of long-term stability in urges to drink is consistent with the assumptions of psychological theories e.g., PRIME theory, which view urges at momentary states [19].

Previous studies have found that the accuracy of the AUDIT questionnaire in discriminating whether one has an alcohol use disorder according to DSM-IV and ICD-10 criteria is as high as 0.9 [43,44]. Literature is however lacking on ROC_AUCs_ for predictors of attempts to cut down and alcohol consumption i.e., actual behaviour change. Studies on cigarette dependence have found that ROC_AUCs_ for attempts to quit smoking are in a similar range to those identified in the current study [21]. This likely reflects greater difficulties in predicting behaviour change due to instability over time. For example, the number of drinks consumed on any one occasion is strongly associated with pre-drinking mood [45].

An additional point of interest is that a significant number of high-risk drinkers attempted to cut down after reporting that they did not have any urges to drink. This provides further evidence that behaviour is a relatively complex and unstable phenomenon and results from the interplay between multiple motivational influences on a moment-to-moment basis e.g., plans, beliefs, views, evaluations, and desires [19]. It also suggests that health-care professionals should not stop encouraging patients to cut down on their alcohol consumption even if they do not report strong desires to drink [46].

## 5. Conclusions

In conclusion, this single item measure of urges to drink may be an efficient quantitative tool for population level surveys and studies assessing the impact of interventions aimed at helping high-risk drinkers reduce their alcohol consumption. The fact that it involves reported experience in the previous 24 h means that it might form a helpful dynamic measure, which is a limitation of the AUDIT questionnaire. Further research should assess the external validity of this measure in different populations and examine short-term test-retest reliability.

## Figures and Tables

**Figure 1 ijerph-16-03714-f001:**
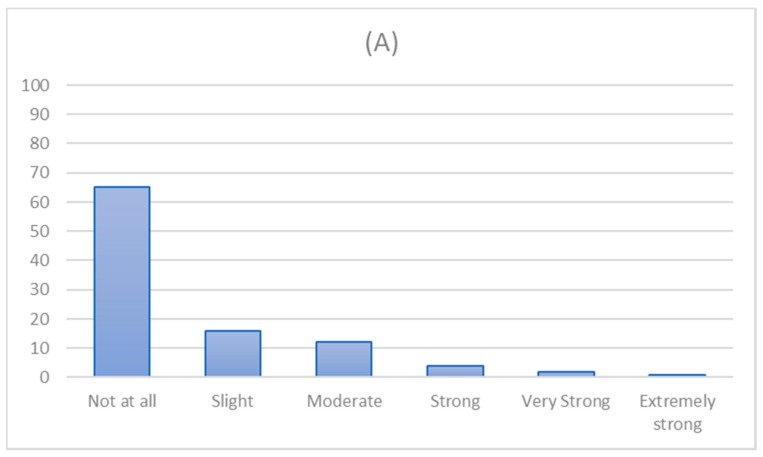
Distribution of scores on SUTD scale at (**A**) baseline and (**B**) follow-up among those completing the 6 month follow-up (*n* = 2960).

**Figure 2 ijerph-16-03714-f002:**
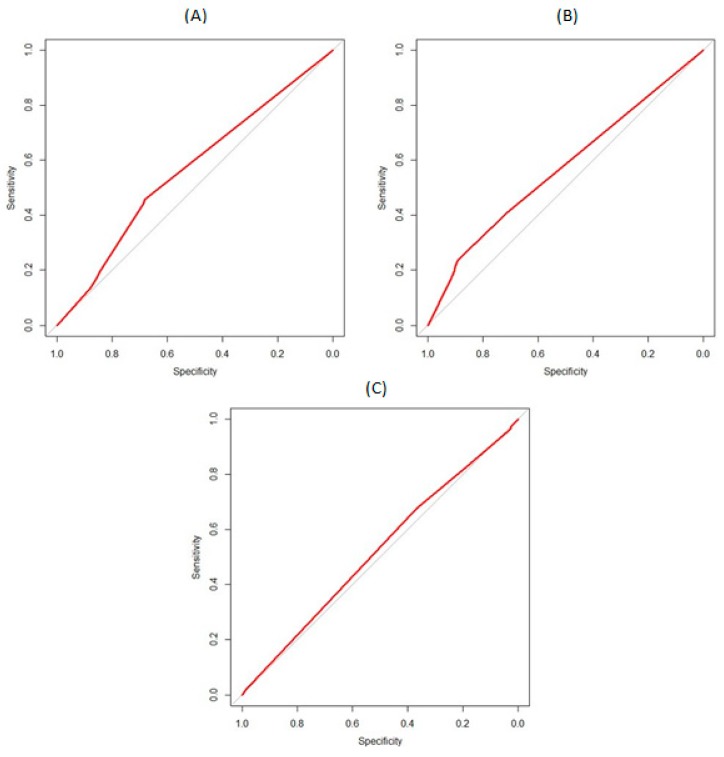
Receiver operating characteristic (ROC) curves showing the accuracy of the SUTD in predicting (**A**) attempts to cut down between baseline and 6-month follow-up [Area under the ROC curve = 0.6]; (**B**) alcohol consumption at 6-month follow-up [Area under the ROC curve = 0.6]; and (**C**) change in alcohol consumption between baseline and 6 months follow-up [Area under the ROC curve = 0.5].

**Table 1 ijerph-16-03714-t001:** Participants characteristics overall and as a function of whether they were followed up.

	All High-Risk Drinkers (*n* = 14,639)		Followed-Up (*n* = 2960)		Not Followed-Up (*n* = 11,679)		*p*
	%	95% CI	%	95% CI	%	95% CI	
**Gender**							
*Female*	35.1	34.4 to 35.9	35.1	33.4 to 36.9	35.1	34.3 to 36.0	>0.990
*Male*	64.9	64.1 to 65.6	64.9	62.1 to 66.6	64.9	64.0 to 65.7	
**Age**							
16–24	20.5	19.9 to 21.2	11.9	10.8 to 13.2	22.7	21.9 to 23.5	<0.001
25–34	14.5	13.9 to 15.1	9.4	8.4 to 10.5	15.8	15.1 to 16.4	
35–44	14.5	13.9 to 15.0	13.9	12.6 to 15.2	14.6	14.0 to 15.3	
45–54	17.9	17.3 to 18.6	20.5	19.1 to 22.1	17.3	16.6 to 18.0	
55–64	16.7	16.1 to 17.3	21.9	20.5 to 23.5	15.3	14.7 to 16.0	
65+	16.0	15.4 to 16.6	22.3	20.9 to 23.9	14.3	13.7 to 15.0	
**Social grade**							
*ABC1*	62.0	61.2 to 62.8	70.1	68.4 to 71.7	59.9	59.0 to 60.8	<0.001
*C2DE*	38.0	37.2 to 38.8	29.9	28.3 to 31.6	40.1	39.2 to 41.0	
**Educational qualification**							
*Voluntary*	71.9	71.2 to 72.7	74.7	73.1 to 76.2	71.2	70.4 to 72.0	<0.001
*Non-voluntary*	28.1	27.3 to 28.8	71.2	70.4 to 72.0	28.8	29.6 to 28.0	
**Disability**							
*Yes*	8.6	8.1 to 9.0	11.1	10.0 to 12.3	7.9	7.4 to 8.4	<0.001
*No*	91.4	91.0 to 91.9	88.9	87.7 to 90.0	92.1	91.6 to 92.6	
**Ethnicity**							
*White*	95.4	95.1 to 95.8	96.3	95.6 to 97.0	95.2	94.8 to 95.6	0.011
*Other*	4.6	4.9 to 4.2	3.7	3.0 to 4.4	4.8	4.4 to 5.2	
**Government Office Region**							
*North*	58.6	57.8 to 59.4	57.5	55.7 to 59.3	58.9	58.0 to 59.8	0.153
*South*	41.4	40.6 to 42.2	42.5	40.7 to 44.3	41.4	40.2 to 40.2	
*Urges to drink (baseline)*							
*Not at all*	64.1	63.4 to 64.9	64.7	63.0 to 66.4	64.0	63.1 to 64.9	0.382
*Slight*	17.1	16.5 to 17.7	17.0	15.7 to 18.4	17.1	16.4 to 17.8	
*Moderate*	12.9	12.3 to 13.4	13.2	12 to 14.5	12.8	12.2 to 13.4	
*Strong*	3.5	3.2 to 3.8	2.9	2.3 to 2.6	3.6	3.3 to 4.0	
*Very strong*	1.4	1.3 to 1.6	1.3	0.9 to 1.8	1.5	1.3 to 1.7	
*Extremely strong*	0.9	0.7 to 1.0	0.8	0.5 to 1.2	0.9	0.7 to 1.1	
**Cutting down (baseline)**							
*Yes*	19.8	19.1 to 20.4	23.4	21.9 to 25.0	18.9	18.2 to 19.6	
*No*	80.2	79.6 to 80.9	76.6	75.0 to 78.1	81.1	80.4 to 81.8	<0.001
	**Mean**	**95% CI**	**Mean**	**95% CI**	**Mean**	**95% CI**	
**AUDIT-score (baseline)**	8.5	8.4 to 8.6	8.3	8.2 to 8.5	8.5	8.5 to 8.6	0.218

Note: *p*-value derived from chi-square test for categorical data, Mann–Whitney U for ordinal data and *t*-test for continuous data; AUDIT = Alcohol Use Disorders Identification Test.

**Table 2 ijerph-16-03714-t002:** Results of the regression analysis assessing the association between attempts to cut down drinking between baseline and 6-month follow-up (any versus none), mean consumption at follow-up and mean change in consumption between baseline and 6-month follow-up with the SUTD scale (*N* = 2960).

**Levels of Urges to Drink at Baseline**	**Attempt to Cut Down at Follow-Up % (*n*)**		**OR**	**95% CI**		**Adjusted OR**	**95% CI**	
**SUTD**	**No**	**Yes**		**Lower**	**Upper**		**Lower**	**Upper**
Not at all	78.8 (1510)	21.2 (406)						
Slight	65.7 (331)	34.3 (173)	1.94	1.57 ***	2.41	1.78 ***	1.43	2.22
Moderate	65.7 (257)	34.3 (134)	1.94	1.53 ***	2.45	1.54 ***	1.20	1.96
Strong	65.1 (56)	34.9 (30)	1.99	1.25 **	3.12	1.43	0.87	2.30
Very strong	63.2 (24)	36.8 (14)	2.17	1.09 *	4.18	1.62	0.79	3.22
Extremely strong	62.5 (15)	37.5 (9)	2.23	0.93	5.05	1.16	0.45	2.78
SUTD-3								
Not all	78.8 (1510)	21.2 (406)						
Slight	65.7 (331)	34.3 (173)	1.94 ***	1.57	2.41	1.78 ***	1.43	2.22
>Moderate	65.3 (352)	34.7 (187)	1.98 ***	1.60	2.43	1.51 ***	1.21	1.88
**Levels of Urges to Drink at Baseline**	**Mean (SD) Consumption at Follow-Up**		**Β**	**95% CI**		**Adjusted β**	**95% CI**	
				Lower	Upper		Lower	Upper
Not at all	4.46 (1.48)							
Slight	4.77 (1.39)		0.30 ***	0.16	0.45	0.22 **	0.08	0.35
Moderate	5.23 (1.39)		0.77 ***	0.61	0.93	0.53 ***	0.38	0.69
Strong	5.27 (1.45)		0.80 ***	0.49	1.12	0.40 *	0.09	0.70
Very strong	5.03 (1.73)		0.56 *	0.09	1.03	0.19	−0.26	0.64
Extremely strong	5.26 (2.40)		0.80 **	0.20	1.40	−0.22	−0.81	0.36
SUTD-3								
Not all	4.46 (1.48)							
Slight	4.77 (1.39)		0.30 ***	0.16	0.45	0.22 **	0.08	0.35
>Moderate	5.22 (1.48)		0.78 ***	0.62	0.90	0.46 ***	0.32	0.60
**Levels of Urges to Drink at Baseline**	**Change Consumption (Follow-Up—Baseline) ^a^**		**Β**	**95% CI**		**Adjusted β**	**95% CI**	
				Lower	Upper		Lower	Upper
Not at all	0.39 (1.51)							
Slight	0.24 (1.43)		−0.14	−0.29	0.00	−0.16 *	−0.31	−0.02
Moderate	0.06 (1.43)		−0.33 ***	−0.49	−0.16	−0.40 ***	−0.56	−0.24
Strong	0.22 (1.25)		−0.17	−0.49	0.16	−0.37 *	−0.69	−0.05
Very strong	0.29 (1.80)		−0.10	−0.58	0.38	−0.36	−0.84	0.12
Extremely strong	1.43 (2.43)		1.05 ***	0.44	1.66	0.47	−0.15	1.09
SUTD-3								
Not all	0.39 (1.51)							
Slight	0.24 (1.43)		−0.14	−0.29	0.00	−0.16 *	−0.31	−0.02
>Moderate	0.16 (1.51)		−0.23 **	−0.37	−0.08	−0.36 ***	−0.51	−0.22

Note: OR = odds ratio; OR and β adjusted for age, sex, social grade, region, receipt of a voluntary educational qualification, ethnicity, disability, AUDIT and wave of the survey; * significant at *p* < 0.05; ** significant at *p* < 0.01; *** significant at *p* < 0.001; ^a^ Positive score = higher consumption at follow-up than baseline, negative score = lower consumption at follow-up than baseline; SUTD = Strength of Urges to Drink Scale

**Table 3 ijerph-16-03714-t003:** Results of the regression analysis assessing the association between mean consumption at follow-up and mean change in consumption between baseline and 6-month follow-up with the SUTD scale restricted to participants cutting down at baseline (*N* = 692).

**Levels of Urges to Drink at Baseline**	**Mean (SD) Consumption at Follow-Up**	**Β**	**95% CI**		**Adjusted β**	**95% CI**	
			**Lower**	**Upper**		**Lower**	**Upper**
Not at all	4.74 (1.49)						
Slight	4.91 (1.29)	0.15	−0.14	0.45	0.08	−0.20	0.37
Moderate	5.30 (1.51)	0.26	−0.31	0.83 *	0.32	0.02	0.61
Strong	5.00 (1.33)	0.55 ***	0.25	0.85	−0.12	−0.68	0.45
Very strong	5.00 (2.16)	0.26	−0.56	1.09	−0.14	−0.95	0.67
Extremely strong	4.62 (3.11)	−0.11	−1.16	0.93	−1.21 *	−2.27	−0.15
SUTD−3							
Not all	4.74 (1.49)						
Slight	4.91 (1.29)	0.15	−0.14	0.45	0.08	−0.21	0.37
>Moderate	5.20 (1.63)	0.45 ***	0.19	0.72	0.18	−0.09	0.45
**Levels of Urges to Drink at Baseline**	**Change Consumption (Follow-Up—Baseline) ^a^**	**Β**	**95% CI**		**Adjusted β**	**95% CI**	
			**Lower**	**Upper**		**Lower**	**Upper**
Not at all	0.22 (1.60)						
Slight	−0.01 (1.32)	−0.23	−0.55	0.08	−0.20	−0.51	0.12
Moderate	0.09 (1.60)	−0.13	−0.45	0.19	−0.20	−0.52	0.12
Strong	0.43 (1.20)	0.20	−0.40	0.81	−0.13	−0.75	0.48
Very strong	0.46 (2.11)	0.24	−0.64	1.11	−0.12	−1.00	0.76
Extremely strong	2.75 (3.58)	2.53 ***	1.42	3.63	1.66 **	0.51	2.80
SUTD-3							
Not all	0.22 (1.60)						
Slight	−0.01 (1.32)	−0.23	−0.55	0.08	−0.19	−0.51	0.12
>Moderate	0.30 (1.79)	0.07	−0.22	0.36	−0.13	−0.43	0.16

Note: OR = odds ratio; OR and β adjusted for age, sex, social grade, region, receipt of a voluntary educational qualification, ethnicity, disability, AUDIT and wave of the survey; * significant at *p* < 0.05; ** significant at *p* < 0.01; *** significant at *p* < 0.001; **^a^** Positive score = higher consumption at follow-up than baseline, negative score = lower consumption at follow-up than baseline.

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
