# Peer review of "Predictive Validity, Diagnostic Accuracy and Test-Retest Reliability of the Strength of Urges to Drink (SUTD) Scale"

_ijerph, 2019, doi:10.3390/ijerph16193714_

Round 1

Reviewer 1 Report

Summary In this manuscript, Beard et al. validate the SUTD against the well-established AUDIT questionnaire. The article is well-written and statistically sound. The collective investigated provides reasonable group sizes to test the hypotheses in question and give perspective on the potential benefits and use cases for prediction of patient outcome, thereby enabling proper preventive measures. Also, by including a broader spectrum of addiction, prevention could potentially be extended to drinkers on the verge of addiction, thereby providing a valuable tool for clinicians to counteract earlier in the addiction process.    Minor points 1. The discussion is not precise enough in respect to the comparison of SUTD and AUDIT and leaves the impression of bias in favor of the SUTD (e.g. "Long-term test re-test reliability was fair for the SUTD and moderate for the AUDIT" How does fair relate to moderate?). Please use clear phrasing and comparative language (better/worse) to help the reader evaluate when to use what questionnaire in relation to the different aspects. 2. At what time of the day were the questionnaires requested? If there was no such request or directive, is there a possibility to get the daytime of taking the test from the data? It would be interesting to see whether the urge to drink varies with the time of day the tests were done. Please clarify in the manuscript and analyze, if possible. 3. Please correct the few grammar and spelling mistakes (e.g. “simper” on page 3, “there is clear a“ on page 5, “a efficient” on page 12) 3. Remarks • It is nice to see the manuscript being processed supposedly with Markdown and R, the only downside for Revision was the inconsistent line numbering, starting only on page 10. • the dynamic aspect of the SUTD might be beneficial to App-based automatic support for the prevention of a recidive.

Author Response

Reviewer 1

Thank you for your comments on this manuscript. We have found them extremely helpful and have addressed them below.

The discussion is not precise enough in respect to the comparison of SUTD and AUDIT and leaves the impression of bias in favor of the SUTD (e.g. "Long-term test re-test reliability was fair for the SUTD and moderate for the AUDIT" How does fair relate to moderate?). Please use clear phrasing and comparative language (better/worse) to help the reader evaluate when to use what questionnaire in relation to the different aspects.

The terms moderate and fair come from the standard cut-offs used for Kappa recommended by Cohen. We now reference this as follows in the analysis section: “Cohen suggested the Kappa result be interpreted as follows: values ≤ 0 as indicating no agreement and 0.01–0.20 as none to slight, 0.21–0.40 as fair, 0.41– 0.60 as moderate, 0.61–0.80 as substantial, and 0.81–1.00 as almost perfect agreement (Cohen, 1960).” We have also changed some of the wording in discussion to make this clearer and also included this reference there.

At what time of the day were the questionnaires requested? If there was no such request or directive, is there a possibility to get the daytime of taking the test from the data? It would be interesting to see whether the urge to drink varies with the time of day the tests were done. Please clarify in the manuscript and analyze, if possible.

Interviews can happen at any time of the day but generally morning interviews are avoided to maximise participant availability. We have now included more details on data collection in the design section “The baseline survey uses a type of random location sampling, which is a hybrid between random probability and simple quota sampling. England is first split into 171,356 ‘Output Areas’, comprising of approximately 300 households. These areas are then stratified based on ACORN characteristics and geographic region. ACORN is a socio-economic profiling tool developed by CACI (http://www.caci.co.uk/acorn/). The areas are then randomly allocated to interviewers, who travel to their selected areas and conduct the electronic interviews with one member of the household. Interviews are conducted until quotas based upon factors influencing the probability of being at home and tailored to local area census data are fulfilled. Morning interviews are avoided to maximise participant availability.” We have also added this as a limitation in the discussion “Finally, interviews could happen at any time of the day but morning interviews were avoided to maximise availability. It is possible that urges to drink are different in the morning and evening and that any differences are moderated by dependence levels. Dependent drinkers may have a greater urge for ‘relief drinking’ in the morning, while heavy non-dependent drinkers could be more affected by cues for early evening drinking”.

Please correct the few grammar and spelling mistakes (e.g. “simper” on page 3, “there is clear a“ on page 5, “a efficient” on page 12)

Thank you we have made these corrections

Reviewer 2 Report

Thank you for allowing me to read your paper which I found to be very interesting indeed. Overall, I found the paper to be very well written and well presented and one that makes a useful contribution to the literature on the use of instruments to assess and diagnose potential/actual alcohol problems. My comments are only very minor:

Abstract – The abstract is currently at 246 words and so this may need shortened still. Also, in the fourth sentence the test retest reliability assessment is described as test retest relatability which requires to be amended.

In terms of KEYWORDS – At the top of the abstract, the keywords there include the word alcohol but they don’t in the line below the abstract and so this ought to be made consistent.

There are also an additional couple of typographical errors:

P3, para 1 - urges to smoke is written where should be urges to drink?

Page 5 – Near the beginning of Section 3.2.1. word missing – ‘drinkers’ is missing from ‘high risk’ (drinkers).

Author Response

Thank you for your comments on this manuscript. We have found them extremely helpful and have addressed them below.

Abstract – The abstract is currently at 246 words and so this may need shortened still. Also, in the fourth sentence the test retest reliability assessment is described as test retest relatability which requires to be amended.

Thank you we have corrected this error and reduced the abstract down to 211 words

In terms of KEYWORDS – At the top of the abstract, the keywords there include the word alcohol but they don’t in the line below the abstract and so this ought to be made consistent.

We have deleted the second set of keywords. This was an error.

There are also an additional couple of typographical errors. P3, para 1 - urges to smoke is written where should be urges to drink? Page 5 – Near the beginning of Section 3.2.1. word missing – ‘drinkers’ is missing from ‘high risk’ (drinkers).

We have made these changes.